# Orchard Spray Study: A Prediction Model of Droplet Deposition States on Leaf Surfaces

**Jun Li [1,2], Huajun Cui [1], Yakun Ma [1], Lu Xun [3], Zhiqiang Li [1], Zhou Yang [2,4] and Huazhong Lu [2,5,*]**

[1]   College of Engineering, South China Agricultural University, Guangzhou 510642, China;
     autojunli@scau.edu.cn (J.L.); cuihuajun@stu.scau.edu.cn (H.C.); yakunma@stu.scau.edu.cn (Y.M.);
     zhiqiang.li@stu.scau.edu.cn (Z.L.)

[2]   Key Laboratory of Key Technology on Agricultural Machine and Equipment, Ministry of Education,
     South China Agricultural University, Guangzhou 510642, China; yangzhou@scau.edu.cn

[3]   Department of Agri-Food Engineering and Biotechnology, Universitat Politècnica de Catalunya,
     Parc Mediterrani de la Tecnología, Campus del Baix Llobregat, Esteve Terradas, 8,
     08860 Castelldefels, Barcelona, Spain; lu.xun@upc.edu

[4]   Guangdong Provincial Key Laboratory of Conservation and Precision Utilization of Characteristic
     Agricultural Resources in Mountainous Areas, Jiaying University, Meizhou 514015, China

[5]   Guangdong Academy of Agricultural Sciences, Guangzhou 510640, China

*    Correspondence: huazlu@scau.edu.cn

**Abstract:** During air-assisted spraying operations in orchards, the interaction between the droplets and the target leaves has a decisive influence on the retention of the droplets on the leaves and the final deposition state. Based on the observation of the final deposition effect of the droplets in the spray test, the retention state of the droplets on the leaves is divided into three categories: uniform distribution (hereinafter referred to as uniform), accumulation, and loss. During the initial interaction between the droplets and the leaves, the adhesion or sliding state of the droplets has an important influence on the final deposition state of the droplets, which is determined by the target leaf adhesion work in this paper. Based on obtaining the characteristic parameters of the leaf surface, a theoretical model of adhesion work related to parameters such as the contact angle, rough factor, and initial tilt angle of the leaf is established. Afterward, through the connection of the droplet coverage on the macro level, the establishment of the deposition state model of the droplet group on the leaf is completed. By conducting the experiment test based on the Box-Behnken design of response surface methodology (RSM), the droplet deposition states under the influence of the spray distance, fan outlet wind speed and droplet size were studied and compared with the predicted values. The test results show that the prediction accuracies of the three states of uniform, accumulation, and loss were 87.5%, 80%, and 100%, respectively. The results of the study indicate that the established prediction model can effectively predict the deposition states of droplets on leaves and provide a reference for the selection of spray operation parameters.

**Keywords:** air-assisted spray; leaf characteristics; spray parameters; prediction of deposition state; parameter decision

## 1. Introduction

The application of pesticides is an indispensable preventative measure that helps prevent yield losses due to organisms and pests that are harmful to crops [1]. In the ideal spraying condition of chemical pesticides, a large number of spray droplets can hit and adhere to the leaves surface during spraying [2]; Droplets attached to the surface of the leaves can stay on the leaves stably [3]; and after the water in chemical pesticides evaporates, the active ingredients are left behind [4,5]. The process by

which sediments are deposited on plants and ingested into their leaves is often called absorption [6]. The deposition, wettability, and adhesion behaviors of pesticide spray droplets on leaf surfaces are crucial in plant protection, owing to their potential effectiveness in reducing chemical wastage and environmental pollution. In conventional pesticide spraying, there are millions of droplets of pesticide spray that do not apply to the surface of the plant and enter the non-target area. The droplets were either blown off by the wind or bounced off the leaf during flight [7,8]. Therefore, fixing most droplets on the surface of the target leaves to prevent chemical loss is a problem worthy of attention [9]. This effect is achieved using an air-assisted sprayer, in which sprays generated by a fan carry pesticide droplets to the target canopy. Air-assisted sprayers generate a strong stream of air to blow the secondary atomized liquid to the target via a fan, and the airflow drives the leaves to flip so that both the obverse and the reverse of the leaves can be reached by the liquid, and this process has been widely used in agriculture and horticulture [10,11]. In the study of airborne spray technology, liquid volume ratio, target position, and airflow velocity all have effects on spray deposition and distribution uniformity. The decrease in spray volume ratio will reduce the amount of spray deposition, but improve the distribution uniformity [12]. At present, it is also important to study the precision and high utilization rate of the amount of pesticide sprayed by air-assisted spray. Factors such as the deposition state, wettability, and adhesion behavior of spray droplets on the leaf surface play a crucial role in the study of air-assisted spray.

The pesticide utilization efficiency is influenced by many factors, and the final amount of pesticide that is deposited inside a target tree canopy is influenced by the physical properties of the spray, the sprayer design and settings, the spray operation parameters, the orchard characteristics, and the weather conditions [13,14]. Spray application and efficacy of leaf pesticides depend on four processes, namely, the deposition, retention, uptake, and translocation of the active compounds in the applied formulation [15]. The spray deposition and droplet retention on the leaf surface determine the pesticide utilization efficiency. The bearing capacity of the leaf surface for droplets is especially important in droplet deposition. There is a critical volume of liquid that can be carried by crop leaves, which is called the first-order loss point. There is an automatic loss of fluid when the liquid volume exceeds this value. After this loss, the liquid reaches the maximum stable retention on the leaf surface [16]. Besides, the interaction between spray droplets and leaf characteristics directly affects the retention of spray droplets [17,18] and the droplet deposition [19]. Mathieu Massinon's team used a combination of an orbital injector and a high-speed camera to carry out droplet impact tests on four representative plant leaves with three different reagents and obtained the impact states of three droplets and leaf: adhesion, bounce, and shatter [2]. Whether the result of droplet impact is adhesion, bounce, or shatter to depend on various factors of the surface of the object and the spray droplet [20]. When a droplet impacts a surface, the kinetic energy of the droplet causes the droplet to spread out across the leaf surface, and the surface tension of the droplet causes the droplet to recoil. If the energy loss is low enough, the droplet will bounce off the leaf, but if the energy loss is too great, the droplet will adhere to the leaf surface [21]. If a droplet impacts a surface at a high speed, the tension can be insufficient to maintain its integrity, and the droplet can shatter into finer droplets [22]. In addition, the retention of sprays by leaves can also be affected by plant characteristics, such as the plant and leaf size [23,24] and the low or variable wettability of the leaf surfaces [25]. The hydrophobic characteristics of the leaf surface are one of the important factors leading to the low efficiency of spray droplet deposition. The hydrophilic and hydrophobic characteristics of plant leaves are inseparable from the waxy characteristics and surface microstructure [20,26,27]. When the droplets form a large contact angle hysteresis on the leaf surface, the adhesive force generated by the liquid surface tension will lead to the phenomena of contact line pinning, which inhibits the droplets from sliding on the leaf surface [28]. When the leaf is tilted, the droplets bulge under the action of gravity. The motion of the droplets on the leaf is defined by the balance of forces, including adhesion, gravity, and shear force. For larger droplets, the increase of adhesion is related to the increase of contact area between the droplets and the leaf [28]. To achieve optimal spray retention, more droplets should adhere to the leaf surface [29].

Many studies have also shown that reducing the surface tension of the liquid can reduce the loss of the liquid and increase the liquid deposition on the leaf surface. The common method to change the droplet surface tension is to add surfactant to the liquid to increase the diffusion capacity of liquid droplets on the surface, to increase the liquid deposition on the leaf surface [25,27]. Fengyan Wang's team studied the distribution of oil-soluble surfactant in the suspension after the droplet was dried, and concluded that the higher the oil-soluble surfactant content was, the more widely and evenly the green test emulsion was distributed [6]. However, some studies have shown that although the addition of surfactant to the sprayed liquid can improve the deposition performance of the droplets, it is easy to cause the droplets to become so small that they are prone to drift and evaporation during the spraying process [30]. Another way to improve droplet deposition performance is to mix polymer additives into the sprayed liquid to inhibit droplet shrinkage after hitting the surface, thus preventing the droplet from rebounding [9,29]. Maher Damak's team carried out spray experiments on target leaves by using two spray liquids with opposite charges. When two droplets with opposite charges met on the leaf surface, they formed a hydrophilic defect and adhered to the leaf surface. Hydrophilic defects can make more droplets adhere to the leaf surface [20]. Thus, it can be seen that the physicochemical properties of droplets are one of the important factors in the process of improving the spray effect of air delivery.

To date, numerous scholars have focused on the theory surrounding the droplet-leaf interaction to identify the characteristics of target leaves and the optimal parameters of the air-assisted spray for droplet deposition on leaves [31], and the spray operation parameters can be optimized to suit different characteristics of the target leaves and orchard canopy structures [23,24]. However, research on droplet-leaf interaction has been based on static target leaves [32]. Minimal research has been conducted on dynamic target leaves (The movement of the leaf in a wind-driven state) and spray droplets. In addition, orchard-spray interactions differ from those of field crops. Fine spray droplets mainly adhere to leaf surfaces, and the spray droplet deposition states on orchard leaf surfaces are greatly influenced by droplet-leaf interactions, including uniform, gathering, and loss interactions. This study aimed to investigate the mechanism of the droplet and leaf interaction and the deposition states of spray droplets on leaf surfaces when the spray droplets adhere to a dynamic leaf surface. A deposition state-prediction model was developed for spray droplet deposition on hydrophilic and hydrophobic leaves based on the dynamic interactions between the droplets and the leaves, which revealed the behaviors and energy changes in the interactions between spray droplets and dynamic leaves. The deposition state-prediction model provides a useful reference for the development of optimal spray parameters for an orchard.

## 2. Materials and Methods

### 2.1. The Deposition State of Spray Droplet

Since as early as the 1960s, many people have studied the rolling phenomenon of droplets on leaves. It was found that the droplets are spherical at first, but they are deformed due to the changes of the surface inclination. The greater the inclination, the faster and more violent is the deformation. Additionally, it was found that on surfaces that are difficult to wet, droplets will roll when the droplet size reaches a certain volume. As the amount of spray increases, the droplets become larger. When the gravity of the droplet exceeds the holding force of the leaf on the droplet, the droplet falls from the leaf surface. This phenomenon very easily occurs in conventional thick-fog large-capacity spraying. However, if the leaves are easily wetted or the liquid has too much wetting capacity, the fog droplets can easily spread out on the leaves, becoming a very thin liquid film and staying on the leaves (see Figure 1c). In this situation, if the spray amount is too large, the result is that there cannot be much liquid left on the target surface, as the droplets will slide along the liquid film, the liquid will be lost from the leaf surface, and finally, the deposition amount of droplets will be reduced.

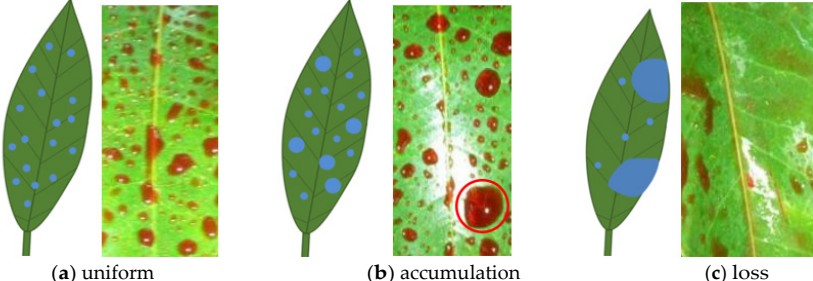

<div align="center">(<b>a</b>) uniform        (<b>b</b>) accumulation        (<b>c</b>) loss</div>

**Figure 1.** Schematic diagram of the deposition states of droplets on the target leaves; (**a**) uniform; (**b**) accumulation; (**c**) loss.

Based on the results of a large number of droplet deposition tests, the deposition states of droplets on target leaves can be divided into the following three types: uniform, accumulation, and loss, as shown in Figure 1. Uniform indicates that most of the droplet sizes deposited on the target leaves are uniform; accumulation indicates that there are obvious large droplets on the target leaves; and loss indicates that the deposited droplets fall from the leaves, resulting in the number of deposited droplets being few. Droplets show different depositional states on the leaves of plants, which are closely related to the interactions between the droplets and the leaves during the deposition process. It can be inferred from the final deposition state that in the uniform state, most of the droplets adhere to the target leaf at the beginning. In the accumulation state, some of the droplets are polymerized with adjacent droplets on the leaf, and small droplets are accumulated into big droplets. In the loss state, each droplet interacts and accumulates into a large droplet. The gravity of the resulting droplets is greater than the adhesion force of the leaves on the droplets, which causes the droplets to slip on the target leaf and finally causes loss. It can be seen that the droplets and the initial interactions of the leaves exhibit adhesion or sliding states, which have an important influence on the final deposition state of the droplet group on the target leaves. To effectively predict the final deposition state of a droplet group on different target leaves, the target leaf adhesion work model is first established to determine the initial interaction between a deposited droplet and the target leaf.

By analyzing the droplets in a spray experiment, we can find the different manifestations of single deposited droplets on the leaf surface. Under a static condition of the leaves, the deposition state of a single deposited droplet on the leaf surface is shown in Figure 2. By observing the uniform state of Figure 1a, it can be found that a large number of deposited droplets present the state shown in Figure 2a,b, and the boundary points M and N remain unchanged. By observing the accumulation state of Figure 1b, it can be found that the presence of large droplets is shown in Figure 2c. Due to the droplets being too large in this case, the droplets overflow the lower boundary point N, and the upper boundary point M does not move. Therefore, the contact surface between the droplet and the leaf surface increases, and the holding force increases, so that the droplet still stays on the leaf, but it will retain the risk of slipping off easily. Observe the state of Figure 1c and the deposited droplets present in Figure 2d. The droplets have slipped out of the initial region and fallen from the leaf.

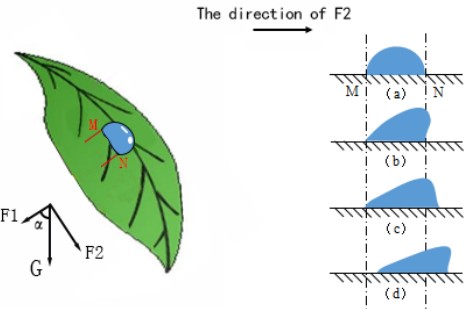

**Figure 2.** Schematic diagram of individual droplet deposition states; (**a**) and (**b**) uniform; (**c**) accumulation; (**d**) loss.

### 2.2. The Parameter Acquisition Test of the Target Leaf Adhesion Work Model

According to previous studies, the surface properties of the target leaves and the physicochemical properties of the sprayed pesticides have significant effects on the adhesion work. The solution used in this experiment was Ponceaus 2R solution (mass ratio: 1:600), and the focus was on the effects of the surface properties of the target leaf on the adhesion work. In this paper, the characteristic parameters of different target leaves are measured, including the rough factor r, contact angle θ, the surface tension of the liquid $\sigma_{lg}$ and critical sliding volume of the droplet under different leaf inclination angles α. Finally, the adhesion work model of the target leaf is established by the linear fitting.

#### 2.2.1. Experimental Materials

The test instrumentation in the laboratory trial mainly included: a Data Physics OCA40 Micro Surface Contact Angle Meter (Germany Dataphysics Co., Ltd., Filderstadt, Germany), an Rtec Instrumentation 3D Profilometer (Germany BMT Co., Ltd., Reichenbach, Germany), and a micropipettor (LiChen Technology Co., Ltd., Shanghai, China). This experiment selected a variety of different characteristics of leaves for relation to the deposition of droplets. The target leaves included Citrus shatangju, Feizixiao Litchi, Longan, and *Psidium guajava L.* All the plants were rooted in pots, and fully expanded and visually healthy leaves were selected for the tests. The surface tension of the Ponceaus 2R solution was measured by the hanging-drop method. The OCA40 Micro Surface Contact Angle Meter was used to automatically record and measure droplet profile changes. The test was conducted 5 times, and the test results were averaged.

#### 2.2.2. Leaf Surface Structure Measurement

Fully expanded, visually healthy leaves were selected from four target plants. The leaf samples were placed on the working stage of the profilometer. Then, the 3D structure of the leaf surface was observed under a 20 × 0.45 lens. The angle of view and distance of the camera were adjusted to keep the leaf in the frame. When the image was clear, the leaf was scanned. The rough factor was defined as the ratio of the actual leaf area to the projected leaf area. Three different areas (1 cm × 1 cm) of each target leaf were randomly selected for measurement. Each target leaf (including the obverse and reverse sides) was scanned three times. The mean of the rough factor was taken.

#### 2.2.3. Contact Angle Measurement

The contact angle (also known as the wetting angle) is defined as the junction of solids, liquids, and gases. The angle between the solid-liquid interface and the gas-liquid interface is usually expressed as the intrinsic contact angle θ of a solid surface. If the droplet spreads, wetting a large area of the surface, then the contact angle θ is less than 90 degrees, and the surface is considered hydrophilic, or water-loving. In contrast, if the droplet forms a sphere that barely touches the surface—like drops of water on a hot griddle—the contact angle θ is more than 90 degrees, and the surface is hydrophobic, or water-fearing [33]. The static contact angle (θ) was measured by the sessile drop method with an OCA40 Micro Contact Angle Meter and a digital camera [34]. Every target leaf (including the obverse and reverse sides) was measured three times to calculate the average.

#### 2.2.4. Critical Sliding Volume Test

The instruments and materials used in this test mainly included the test platform of the air-assisted sprayer, a micropipettor (LiChen Technology Co., Ltd., Shanghai, China) and living plants. Ponceaus 2R Biological Dye solution was again selected as the observation reagent, and the concentration was the same as that of the spray solution used in the leaf characteristics test.

The target leaf was tested in the non-picked form to reduce the potential error caused by the test order. The height of the triangle bracket was adjusted relative to the target leaf, and then the double-sided tape was used to flatly adhere to the target leaf surface to the stage of the tripod bracket.

The angle of the stage was adjustable. A micropipettor was used to drop Ponceau 2R solution onto the leaf surface. To avoid interference in the droplet deposition, the droplets were arranged in a crisscross pattern including 3–6 different volumes of droplets, and the droplet volume was increased from top to bottom and from left to right (Figure 3). After the droplet deposition state was stable (about 10 s), the deposition states of the droplets on the obverse side of the target leaf were observed. A sponge was used to absorb the solution on the leaf surface after every treatment, and the tilt angle of the stage was then adjusted for the next set of tests. The above steps were repeated 6 times for each leaf.

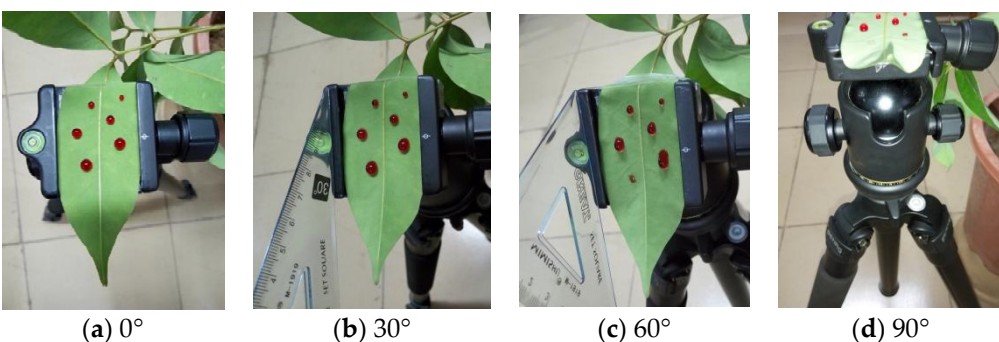

| **(a)** 0° | **(b)** 30° | **(c)** 60° | **(d)** 90° |

**Figure 3.** Measurement of the droplet deposition status and angle of the target-leaf surface; (**a**) Leaf inclination is 0°; (**b**) Leaf inclination is 30°; (**c**) Leaf inclination is 60°; (**d**) Leaf inclination is 90°.

### 2.3. Establishment of the Prediction Model

#### 2.3.1. Establishment of the Target Leaf Adhesion Work Model

In this experiment, the target leaf is stationary; therefore, the initial kinetic energy of the droplet is zero. The droplet gravitational potential energy $E_p = V\rho gh$ (the droplet-surface energy $E_{suf} = \pi D^2 \sigma_{lg}$ ($\times 10^{-8}$ J) is too small to be considered). So, the droplet is mainly governed by gravity and viscosity. When the droplet reaches the maximum adhesion of the leaf, the droplet slides from the leaf to the ground, and the droplet kinetic energy is $E_k = \frac{1}{2}m_0 v_0^2$ when it lands. By analyzing the energy conservation of the whole droplet process, the equation can be obtained:

$$E_p - W_s = \frac{1}{2}m_0 v_0^2 - 0 \tag{1}$$

The adhesion work of droplets on leaf surface can be obtained from Equation (1), that is:

$$W_s = \frac{4}{3}\pi\rho gh D^{*3} - \frac{1}{2}m_0 v_0^2 \tag{2}$$

where $W_s$ is the droplet slide work ($\times 10^{-8}$ J), $D^*$ is the critical diameter of the droplet sliding on the leaf surface (µm), $\rho$ is the droplet density (kg·$m^{-3}$), $g$ (= 9.8 N·$kg^{-1}$) is the acceleration of gravity, h is the height of the leaf from the ground (m), $m_0$ is the mass of the droplet when it falls to the ground and $v_0$ is the instantaneous velocity of the droplet when it falls to the ground.

Multiple regression equations for the droplet slide work, contact angle and rough factor were established using multiple stepwise regression analysis:

$$W_s = 10^{-6}\left(21.8\theta + 1600r - 6.25\alpha - 500r^2 + 1400\sigma_{lg} - 1800\right) \tag{3}$$

where $W_s$ is the slide work ($\times 10^{-8}$ J); $r$ is the leaf surface rough factor ($r \geq 1$); $\theta$ is the static contact angle of the droplet on the leaf surface (°); $\alpha$ is the leaf inclination angle (°) and $\sigma_{lg}$ is the surface tension of the liquid (N/m).

The correlation coefficient of the slide work regression model for the leaf surface is 0.917. Therefore, the slide work regression model of the droplet deposited on the leaf is capable of predicting the slide work of droplets on leaf surfaces.

### 2.3.2. Establishment of the Critical Sliding Particle Size Model of Droplets

In the interactions between droplets and leaf surfaces, the relative velocity of a droplet adhered to a leaf surface is zero. Thus, the droplet adhered to the leaf surface mainly has kinetic energy ($E_{dyn} = \frac{1}{2}mv^2$). The critical diameter of a droplet sliding on the leaf surface was calculated using $E_{com} = 0$, that is:

$$E_{com} = W_s - E_{dyn} = 0 \tag{4}$$

By bringing the adhesion work (Equation (3)) into Equation (4), the following equation can be obtained:

$$D^* = \left[ \frac{3 \times 10^{-6} \left( 21.8\theta + 1600r - 6.25\alpha - 500r^2 + 1400\sigma_{lg} - 1800 \right)}{2\pi\rho v^3} \right]^{\frac{1}{3}} \tag{5}$$

where $D^*$ is the critical diameter of the droplet sliding on the leaf surface (μm), $r$ is the leaf surface rough ratio ($r \geq 1$), $\theta$ is the static contact angle of the droplet on the leaf surface (), $v$ is the value of the target-leaf aerodynamic velocity (m·s$^{-1}$), ρ is the density of the spray liquid (kg·m$^{-3}$), $\alpha$ is the leaf inclination angle (°), and $\sigma_{lg}$ is the surface tension of the liquid (N/m).

### 2.3.3. Prediction Model of the Droplet Deposition State

The two extreme cases of critical sliding droplet formation are shown in Figure 4. The critical sliding droplets shown in Figure 4a consist only of droplets deposited at point P. The critical sliding droplets shown in Figure 4b consist of a spray droplet at point Q and a neighboring spray droplet (n − 1) around point Q. In the case of Figure 4a, the corresponding droplet coverage is generally 100%, and the droplets accumulate and have the risk of slipping off easily. Now, the coverage in the case of Figure 4b can be analyzed. First, let n = D$^*$/d, where D$^*$ and d are the critical diameter of the droplet sliding on the leaf surface and the spray droplet diameter, respectively. In this case, n droplets are closely arranged, the total area of droplet deposition is approximately (n$\pi$d$^2$/4), the area of critical sliding droplet deposition is approximately ($\pi$D$^{*2}$/4), and the ratio of the two deposition areas is nd/D$^{*2}$, which can be taken as the coverage ratio of an individual droplet. Therefore, we substitute n = D$^*$/d into nd/D$^{*2}$, which simplifies to d/D$^*$.

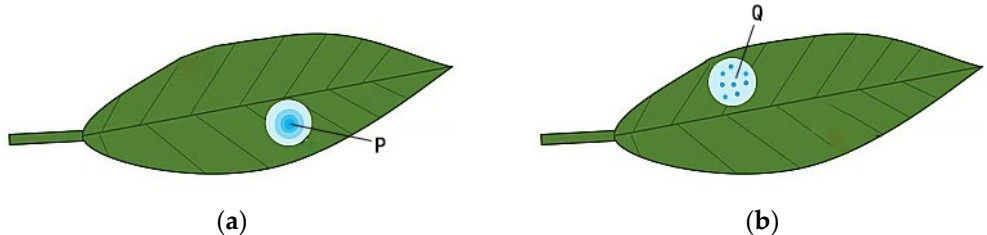

**　　　　　　　　(a)　　　　　　　　　　　　　　　　　　　(b)**

**Figure 4.** Schematic diagram of critical sliding droplet formation; (**a**) The critical sliding droplet; (**b**) The deposited droplet.

Upon combining the two extreme cases of critical sliding droplet formation, the linear interpolation method can be used to obtain the relationship between the droplet coverage C$_{dro}$ and the number of deposited droplets at any point on the leaf surface i:

$$i = 1 + \frac{n-1}{1 - \frac{1}{n}} \left( C_{dro} - \frac{1}{n} \right) \tag{6}$$

where the simplified formula is

$$i = \frac{D^*}{d} C_{dro} \tag{7}$$

Therefore, the diameter of the deposited droplet ($D$) corresponding to the droplet coverage ($C_{dro}$) at this point is approximately $D = id$, where d is the spray droplet diameter. The simplified formula is

$$D = D^* C_{dro} \tag{8}$$

where D is the critical diameter of the droplet. Thus, the kinetic energy of the droplet on the leaf surface is:

$$E_{dyn} = \frac{1}{2}mv^2 = \frac{\pi \rho D^3 v^2}{12} \tag{9}$$

When the kinetic energy of the droplet ($E_{dyn}$) reaches a certain threshold, the droplet will present the state of sliding on the leaf surface. The diameter of the deposited droplet ($D$) corresponding to the sliding state of the droplet and the target-leaf aerodynamic velocity are large when the droplet is present in the state of sliding on the leaf surface. The kinetic energy of the droplet on the leaf surface ($E_{dyn}$) is related to $D$ and $v$; therefore, the minimum kinetic energy of the droplet sliding on the leaf surface is selected as the fitting parameter, and linear regression analysis can be used to develop the relationship between the slide work ($W_s$) and the kinetic energy of the droplet ($E_{dyn}$).

$$W_s = \left(0.87 + 1.56 E_{dyn}\right), \ R^2 = 0.93 \tag{10}$$

In conclusion, the following conditions should be met to predict the loss of droplets on the leaf surface:

$$\begin{cases} \frac{d}{D^*} < C_{dro} < 1 \\ W_s \leq \left(0.87 + 1.56 E_{dyn}\right) \end{cases} \tag{11}$$

The droplet state of accumulation on the leaf surface must meet the following condition:

$$\begin{cases} \frac{d}{D^*} < C_{dro} < 1 \\ W_s \geq \left(0.87 + 1.56 E_{dyn}\right) \end{cases} \tag{12}$$

The droplet state of loss on the leaf surface must meet the following condition:

$$C_{dro} \leq \frac{d}{D^*} \tag{13}$$

In the above formulas, d is the spray droplet diameter ($\mu m$), $D^*$ is the critical size of a droplet sliding on the leaf surface ($\mu m$), $C_{dro}$ is the spray coverage (%), $W_s$ is the adhesion work of a hydrophilic leaf and hydrophobic leaf ($\times 10^{-4}$J), and $E_{dyn}$ is the kinetic energy of a droplet ($\times 10^{-8}$J). The prediction states and test states of the droplets and target leaf interactions are shown in Tables 1–4. If the diameter of the deposited droplet ($D$) is less than the spray droplet diameter (d), the kinetic energy of the droplet can be calculated according to the d value.

**Table 1.** Predicted and actual states of deposition of droplets on the obverse of Litchi leaves under different combinations of operating parameters.

| Test Number | S (m) | $v_{few}$ (m·s$^{-1}$) | d (10$^{-4}$ m) | $D^*$ (10$^{-4}$ m) | D (10$^{-4}$ m) | v (m·s$^{-1}$) | $E_{dyn}$ (10$^{-11}$ J) | d/$D^*$ (%) | $C_{dro}$ (%) | Predicted Result | Experimental Result |
|---|---|---|---|---|---|---|---|---|---|---|---|
| 1 | 1.0 | 7.8 | 4.14 | 35.0 | 8.8 | 0.12 | 0.26 | 11.8 | 25.1 | accumulation | accumulation |
| 2 | 1.0 | 11.5 | 3.42 | 35.0 | 3.78 | 0.22 | 0.07 | 9.77 | 10.8 | accumulation | accumulation |
| 3 | 1.6 | 0 | 4.14 | 43.5 | 4.14 | 0 | 0 | 9.52 | 4.1 | uniform | uniform |
| 4 | 0.4 | 7.8 | 4.80 | 35.0 | 9.27 | 0.19 | 0.75 | 13.71 | 26.5 | accumulation | accumulation |
| 5 | 1.0 | 0 | 3.42 | 43.5 | 3.96 | 0 | 0 | 7.86 | 9.1 | accumulation | uniform |
| 6 | 0.4 | 0 | 4.14 | 43.5 | 10.54 | 0 | 0 | 9.52 | 24.2 | accumulation | accumulation |
| 7 | 1.6 | 7.8 | 4.80 | 35.0 | 6.13 | 0.05 | 0.02 | 13.71 | 17.5 | accumulation | uniform |
| 8 | 0.4 | 7.8 | 3.42 | 35.0 | 8.54 | 0.19 | 0.59 | 9.77 | 24.4 | accumulation | accumulation |
| 9 | 1.0 | 11.5 | 4.80 | 35.0 | 5.53 | 0.22 | 0.21 | 13.71 | 15.8 | accumulation | accumulation |
| 10 | 1.6 | 11.5 | 4.14 | 35.0 | 4.14 | 0.13 | 0.03 | 11.83 | 9.8 | uniform | uniform |
| 11 | 1.6 | 7.8 | 3.42 | 35.0 | 3.42 | 0.05 | 0.003 | 9.77 | 1.4 | uniform | uniform |
| 12 | 0.4 | 11.5 | 4.14 | 35.0 | 7.01 | 0.31 | 0.87 | 11.83 | 20.04 | accumulation | loss |
| 13 | 1.0 | 0 | 4.80 | 43.5 | 11.70 | 0 | 0 | 11.03 | 26.9 | accumulation | accumulation |

The 'D*' is the Critical diameter of the droplet slide on the leaf surface. And the 'd/D*' is the rate of 'd' to 'D*'.

**Table 2.** Predicted and actual states of deposition of droplets on the reverse of Litchi leaves under different combinations of operating parameters.

| Test Number | S (m) | $v_{few}$ (m·s$^{-1}$) | d (10$^{-4}$ m) | $D^*$ (10$^{-4}$ m) | D (10$^{-4}$ m) | v (m·s$^{-1}$) | $E_{dyn}$ (10$^{-11}$ J) | d/$D^*$ (%) | $C_{dro}$ (%) | Predicted Result | Experimental Result |
|---|---|---|---|---|---|---|---|---|---|---|---|
| 1 | 1.0 | 7.8 | 4.14 | 28.3 | 5.93 | 0.12 | 0.08 | 14.63 | 20.94 | Accumulation | Accumulation |
| 2 | 1.0 | 11.5 | 3.42 | 28.3 | 3.42 | 0.22 | 0.05 | 12.08 | 9.50 | Uniform | Uniform |
| 3 | 1.6 | 0 | 4.14 | 39.0 | 4.14 | 0 | 0 | 10.62 | 2.24 | Uniform | Uniform |
| 4 | 0.4 | 7.8 | 4.80 | 28.3 | 5.20 | 0.19 | 0.13 | 16.96 | 18.35 | Accumulation | Accumulation |
| 5 | 1.0 | 0 | 3.42 | 39.0 | 3.42 | 0 | 0 | 8.77 | 5.80 | Uniform | Uniform |
| 6 | 0.4 | 0 | 4.14 | 39.0 | 9.65 | 0 | 0 | 10.62 | 24.75 | Accumulation | Accumulation |
| 7 | 1.6 | 7.8 | 4.80 | 28.3 | 4.80 | 0.05 | 0.007 | 16.96 | 9.68 | Uniform | Uniform |
| 8 | 0.4 | 7.8 | 3.42 | 28.3 | 5.19 | 0.19 | 0.13 | 12.08 | 18.33 | Accumulation | Accumulation |
| 9 | 1.0 | 11.5 | 4.80 | 28.3 | 5.26 | 0.22 | 0.18 | 16.96 | 18.59 | Accumulation | Accumulation |
| 10 | 1.6 | 11.5 | 4.14 | 28.3 | 4.14 | 0.13 | 0.03 | 14.63 | 6.92 | Uniform | Uniform |
| 11 | 1.6 | 7.8 | 3.42 | 28.3 | 3.42 | 0.05 | 0.003 | 12.08 | 1.30 | Uniform | Uniform |
| 12 | 0.4 | 11.5 | 4.14 | 28.3 | 4.14 | 0.31 | 0.18 | 14.63 | 14.51 | Uniform | Loss |
| 13 | 1.0 | 0 | 4.80 | 39.0 | 10.11 | 0 | 0 | 12.31 | 25.93 | Accumulation | Accumulation |

The 'D*' is the Critical diameter of the droplet slide on the leaf surface. And the 'd/D*' is the rate of 'd' to 'D*'.

**Table 3.** Predicted and actual states of deposition of droplets on the obverse of Citrus leaves under different combinations of operating parameters.

| Test Number | S (m) | $v_{few}$ (m·s$^{-1}$) | d (10$^{-4}$ m) | D* (10$^{-4}$ m) | D (10$^{-4}$ m) | v (m·s$^{-1}$) | $E_{dyn}$ (10$^{-11}$ J) | d/D* (%) | $C_{dro}$ (%) | Predicted Result | Experimental Result |
|---|---|---|---|---|---|---|---|---|---|---|---|
| 1 | 1.0 | 7.8 | 4.14 | 29.9 | 5.48 | 0.09 | 0.03 | 13.85 | 18.32 | Accumulation | Accumulation |
| 2 | 1.0 | 11.5 | 3.42 | 29.9 | 4.73 | 0.17 | 0.08 | 11.44 | 15.81 | Accumulation | Accumulation |
| 3 | 1.6 | 0 | 4.14 | 37.0 | 4.14 | 0 | 0 | 11.19 | 5.80 | Uniform | Uniform |
| 4 | 0.4 | 7.8 | 4.80 | 29.9 | 6.21 | 0.14 | 0.34 | 16.05 | 29.16 | Loss | Loss |
| 5 | 1.0 | 0 | 3.42 | 37.0 | 3.43 | 0 | 0 | 9.24 | 9.28 | Accumulation | Uniform |
| 6 | 0.4 | 0 | 4.14 | 37.0 | 9.85 | 0 | 0 | 11.19 | 26.63 | Accumulation | Accumulation |
| 7 | 1.6 | 7.8 | 4.80 | 29.9 | 5.24 | 0.04 | 0.006 | 16.05 | 17.50 | Accumulation | Accumulation |
| 8 | 0.4 | 7.8 | 3.42 | 29.9 | 10.13 | 0.14 | 0.53 | 11.44 | 33.87 | Loss | Loss |
| 9 | 1.0 | 11.5 | 4.80 | 29.9 | 4.80 | 0.17 | 0.08 | 16.05 | 15.68 | Uniform | Accumulation |
| 10 | 1.6 | 11.5 | 4.14 | 29.9 | 4.14 | 0.10 | 0.02 | 13.85 | 8.57 | Uniform | Uniform |
| 11 | 1.6 | 7.8 | 3.42 | 29.9 | 3.42 | 0.04 | 0.002 | 11.44 | 4.35 | Uniform | Uniform |
| 12 | 0.4 | 11.5 | 4.14 | 29.9 | 8.13 | 0.24 | 0.81 | 13.85 | 27.22 | Loss | Loss |
| 13 | 1.0 | 0 | 4.80 | 37.0 | 7.57 | 0 | 0 | 12.97 | 20.45 | Accumulation | Accumulation |

The 'D*' is the Critical diameter of the droplet slide on the leaf surface. And the 'd/D*' is the rate of 'd' to 'D*'.

**Table 4.** Predicted and actual states of deposition of droplets on the reverse of Citrus leaves under different combinations of operating parameters.

| Test Number | S (m) | $v_{few}$ (m·s$^{-1}$) | d (10$^{-4}$ m) | D* (10$^{-4}$ m) | D (10$^{-4}$ m) | v (m·s$^{-1}$) | $E_{dyn}$ (10$^{-11}$ J) | d/D* (%) | $C_{dro}$ (%) | Predicted Result | Experimental Result |
|---|---|---|---|---|---|---|---|---|---|---|---|
| 1 | 1.0 | 7.8 | 4.14 | 31.0 | 6.05 | 0.09 | 0.05 | 13.35 | 19.5 | Accumulation | Accumulation |
| 2 | 1.0 | 11.5 | 3.42 | 31.0 | 3.60 | 0.17 | 0.04 | 11.03 | 11.6 | Accumulation | Accumulation |
| 3 | 1.6 | 0 | 4.14 | 37.9 | 4.14 | 0 | 0 | 10.92 | 9.25 | Uniform | Uniform |
| 4 | 0.4 | 7.8 | 4.80 | 31.0 | 9.91 | 0.14 | 0.50 | 15.48 | 31.95 | Loss | Loss |
| 5 | 1.0 | 0 | 3.42 | 37.9 | 3.90 | 0 | 0 | 9.02 | 10.3 | Accumulation | Uniform |
| 6 | 0.4 | 0 | 4.14 | 37.9 | 12.43 | 0 | 0 | 10.92 | 32.8 | Accumulation | Accumulation |
| 7 | 1.6 | 7.8 | 4.80 | 31.0 | 5.96 | 0.04 | 0.009 | 15.48 | 19.20 | Accumulation | Accumulation |
| 8 | 0.4 | 7.8 | 3.42 | 31.0 | 6.61 | 0.14 | 0.15 | 11.03 | 21.29 | Accumulation | Accumulation |
| 9 | 1.0 | 11.5 | 4.80 | 31.0 | 8.80 | 0.17 | 0.52 | 15.48 | 28.36 | Loss | Loss |
| 10 | 1.6 | 11.5 | 4.14 | 31.0 | 5.52 | 0.10 | 0.0004 | 13.35 | 17.80 | Accumulation | Uniform |
| 11 | 1.6 | 7.8 | 3.42 | 31.0 | 3.42 | 0.04 | 0.002 | 11.03 | 7.69 | Uniform | Uniform |
| 12 | 0.4 | 11.5 | 4.14 | 31.0 | 7.45 | 0.24 | 0.62 | 13.35 | 24.03 | Loss | Loss |
| 13 | 1.0 | 0 | 4.80 | 37.9 | 8.58 | 0 | 0 | 12.66 | 22.65 | Accumulation | Accumulation |

Note: The initial inclination angle of the test leaf is 60°. Under the condition of wind feeding, the leaf angle is calculated at 90°.

### 2.4. Determination Test of the Droplet Deposition State

#### 2.4.1. Materials and Methods

A test platform for the air-assisted sprayer (South China Agricultural University, Guangzhou, China) was used in this study (Figure 5). The spraying test platform height was 1.18 m, and it was equipped with a QiNuo SFWL3-2 axial-flow fan (Hangzhou QiNuo Electromechanical Equipment Co., Ltd., Hangzhou, China). The impeller diameter was 0.36 m, and the height of the fan center relative to the ground was 0.97 m. The fan speed could be set to any speed between 0 and 2800 r/min by adjusting the frequency converter. The detailed parameters are shown as follows: the rated rotation speed was 2800 r/min. The main parts of the spray test platform can be categorized as part of either the spray flow control system or the air-assisted system. The spray flow control system mainly included a pesticide filter, a high-pressure diaphragm pump, an energy accumulator, an overflow valve, a pressure transmitter, a turbine flowmeter, a high-speed solenoid valve, a relay with a modular control panel, an intelligence flow totalizer and a standard spray nozzle.

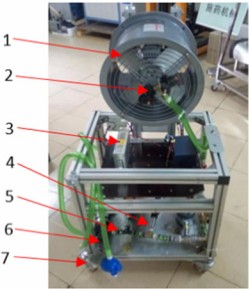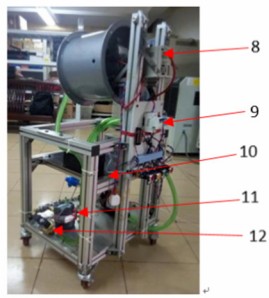

**Figure 5.** Structure of the spraying test platform: (1) airflow fan, (2) nozzle, (3) power switch, (4) puma, (5) filter, (6) adjustable relief valve, (7) manometer, (8) frequency converter, (9) fan switch, (10) intelligence flow totalizer, (11) flowmeter, and (12) solenoid valve.

To minimize the effects of the external environment, the experiment was performed indoors. A test platform consisting of an air-assisted sprayer was used. The experiment was conducted on the leaves of living Citrus plants and Litchi plants, and the leaves were not picked to approximate the field conditions. Leaves of average size were selected as the target leaves. The Citrus leaf was 60 mm wide and 30 mm in length, and the Litchi leaf was 90 mm wide and 40 mm length. The height of the target leaf from the ground was h = 1 m. The Ponceau 2R solution was again used as the spray liquid in the experiment due to its bright color, which made it easy to observe and analyze. The surface tension of the Ponceau 2R obtained through testing was $\sigma_{lg} = 0.071 \, N/m$. The droplet deposition states on the target leaf surfaces were captured by the camera after each spray treatment.

The three flat-fan nozzles were operated at a spray pressure of 0.5 MPa and a spray angle of 80°. The droplet diameters and flow rates of the three types of nozzles were recorded. The image captured by the high-speed camera was processed using ImageJ software to obtain the droplet diameters of the three nozzle types. $D_{V10}, D_{V50}$ and $D_{V90}$ were measured. $D_{V50}$(VMD) was used to describe the spray droplet diameter [35,36]. The measured results of the spray diameter and flow rate for the three nozzles are provided in Table 5. In addition, the airflow velocity (7.8–11.5 m/s) had no significant effect on the secondary atomization of the spray droplets in this study. The influence of the airflow velocity on the droplet diameter was ignored.

**Table 5.** Flow rates and droplet diameters of the three nozzle types.

| Nozzle Type | Flow Rate (L·min$^{-1}$) | Droplet Diameter (μm) |
|:---:|:---:|:---:|
| ATR-RED | 0.69 ± 0.09 | 120 ± 18 |
| ATR-GREEN | 1.39 ± 0.12 | 165 ± 21 |
| ATR-BLUE | 2.31 ± 0.18 | 240 ± 28 |

In this experiment, the laser non-contact dynamic measurement system was used to measure the value of the aerodynamic response velocity of the leaf under the action of the airflow, as shown in Figure 6. A Sunny Instruments laser non-contact dynamic measurement system (Sunny Instruments) was used, with the laser wavelength of the He-Ne laser of 632.8 nm, the working distance of 0.35 m ~ 20 m, and a maximum linear error of 1.00%. The supporting test and analysis software QuickSA2.4.0 was also used. To improve the intensity of the laser reflection signal and ensure the accuracy of the test results, the surface of the target leaf was pasted onto reflective paper, which was lightweight and negligibly affected the quality.

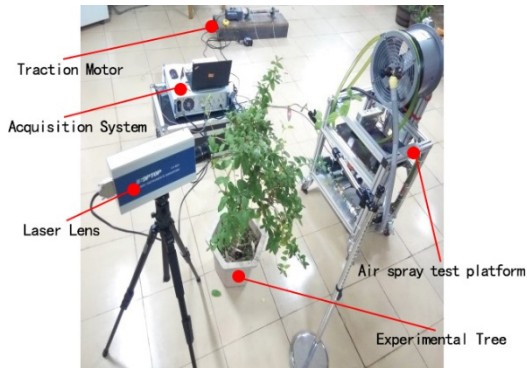

**Figure 6.** Field diagram of the spray test with different operating parameters.

2.4.2. Experimental Procedure

To study the influence of different operational parameters on the actual deposition state of the fog drops and to compare the experimental results with the predicted deposition states to verify the accuracy of the model, an orthogonal experimental design consisting of four influencing factors was carried out. The orthogonal design can greatly reduce the number of tests required and can yield more accurate results.

The spray operation parameters of this experiment included three droplet diameters, three spray distances, three outlet airflow velocities, and four target leaves (Table 6). The programme was designed based on the response surface morphology of the Box–Behnken design. Each target leaf surface was tested with 13 treatments, and each treatment was repeated three times, for a total of 156 trials in this experiment.

**Table 6.** Spray parameters of the trial.

| Spray Parameters | Value |
|---|---|
| Outlet airflow velocity (m.s$^{-1}$) | 0, 7.8, 11.5 |
| Spray distance (m) | 0.4, 1.0, 1.6 |
| Nozzle type | ATR-RED, ATR-GREEN, ATR-BLUE |
| Target leaves | obverse and reverse sides of the Citrus leaf obverse and reverse sides of the Litchi leaf |

The experimental factor level and coding based on Box–Behnken Design RSM methodology [37] are shown in Table 7.

**Table 7.** Test factor level and coding.

| Experimental Factor | Coding and Level | | |
|---|---|---|---|
| | −1 | 0 | +1 |
| Spraying distance (m) | 0.4 | 1.0 | 1.6 |
| Fan outlet wind speed (m.s$^{-1}$) | 0 | 7.8 | 11.5 |
| Particle diameter (μm) | 120 | 165 | 240 |

Different droplet diameters were produced by the three different types of nozzles. The airflow velocity of the fan was regulated by a frequency converter. The spray distance was adjusted by moving the target tree pot position. To reduce the influence of the pressure fluctuation of the experimental platform spraying circuit on the droplet diameter and droplet deposition rate, the test was carried out at a constant temperature of 28 °C and spray pressure of 0.5 MPa. The spray pressure of the test platform was controlled by the overflow value. A pressure gauge monitored the spray pressure in real-time.

The aerodynamic response velocity of the target leaves was measured under three different airflow velocities. Otherwise, the target leaf and axial-flow fan were maintained at the same height in an axial horizontal direction. The target-leaf surface faced the fan. The drive direction of the test platform was kept constant, and the speed was 0.2 m·s$^{-1}$. The branch of the target leaf was fixed with a clamping bracket to prevent the branch shaking from interfering with the test results. The aerodynamic response velocity of the target leaf was measured and recorded by the contactless dynamic measurement system under three airflow velocities.

After each spray test, the camera was used to capture the target leaf droplet deposition image. After the end of the test, the screenshot tool was used to randomly select the middle part of the target leaf surface in the captured droplet deposition image as a droplet coverage analysis object. The images of the droplet deposition states were processed using MATLAB, which could determine the spray coverage of the target leaf. The ratio of the number of pixels occupied by the red fog droplets in the captured image to the total number of pixels in the image was taken as the result of the droplet coverage rate $C_{dro}$ [38]. The target leaf surface droplet deposition state was recorded, and the surface droplet drop coverage was calculated. The test results of the droplet deposition state on the target blade surface are shown in Tables 1–4.

## 3. Results and Discussion

### 3.1. Model Parameters

#### 3.1.1. Rough Factor

The rough factor calculation results are shown in Table 8. The images of the 3D structures show that there were large differences among the surface profiles of the leaves of Citrus, Litchi, Longan, and *Psidium guajava L.* (Figures 7–9). The 3D surface structure for the rough factor of the leaf surface is shown for the range of 1.0~1.3 (Figure 7). The Citrus leaf surfaces (obverse and reverse) contained many micro-cylinders and stoma, were covered with a waxy cuticle and were smooth to the touch. The obverse sides of the Litchi and *Psidium guajava L.* leaves all had folds, and the leaf surfaces were not flat but were smooth.

**Table 8.** The test results of the target leaf surface rough factor.

| Leaf Surfaces | Plants | | | |
|---|---|---|---|---|
| | Citrus | Litchi | Longan | *Psidium guajava* L. |
| Obverse of leaf | 1.13 ± 0.06a | 1.17 ± 0.04b | 1.15 ± 0.03b | 1.32 ± 0.06b |
| Reverse of leaf | 1.18 ± 0.05a | 1.45 ± 0.06a | 1.81 ± 0.05a | 1.92 ± 0.07a |

Note: The measured rough factor consists of the three-replicate mean and standard deviation. Different lowercase letters indicate significant differences ($p < 0.05$).

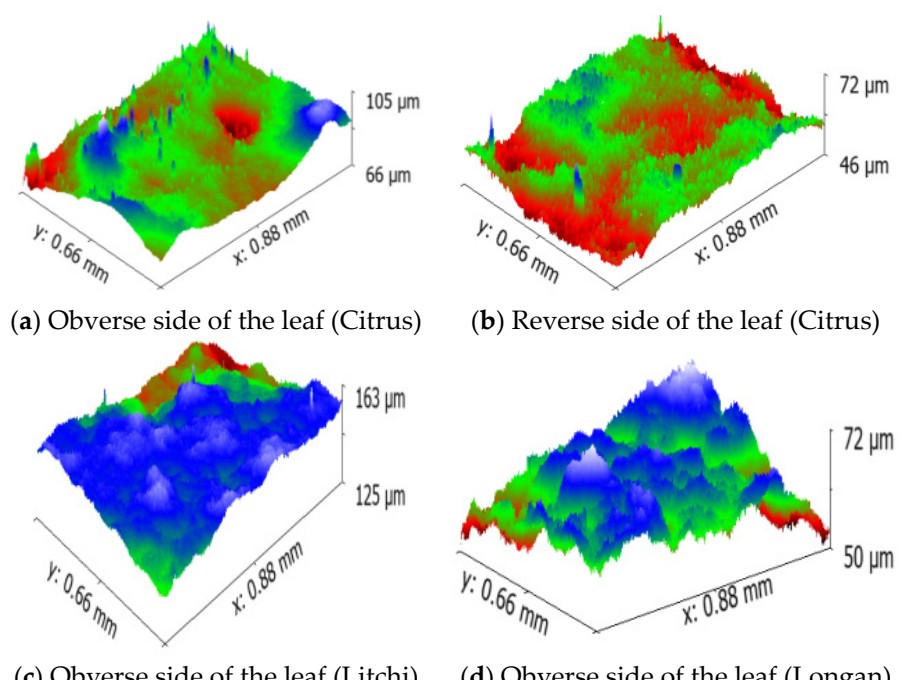

(**a**) Obverse side of the leaf (Citrus)     (**b**) Reverse side of the leaf (Citrus)

(**c**) Obverse side of the leaf (Litchi)     (**d**) Obverse side of the leaf (Longan)

**Figure 7.** Target leaf surface 3D structures for rough factors in the range of 1.0–1.3.

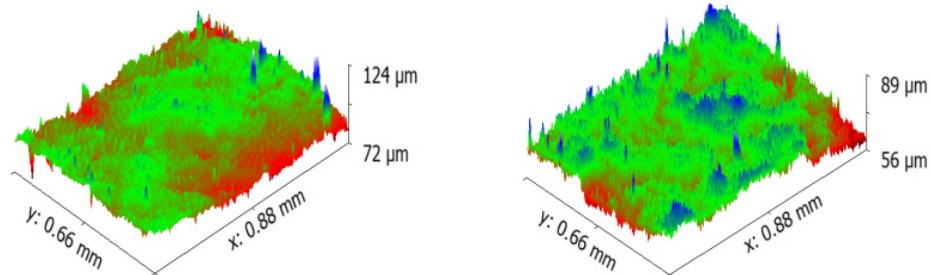

(**a**) Obverse side of the leaf (*Psidium guajava L.*)     (**b**) Reverse side of the leaf (Citrus)

**Figure 8.** Target leaf surface 3D structures for rough factors in the range of 1.3–1.5.

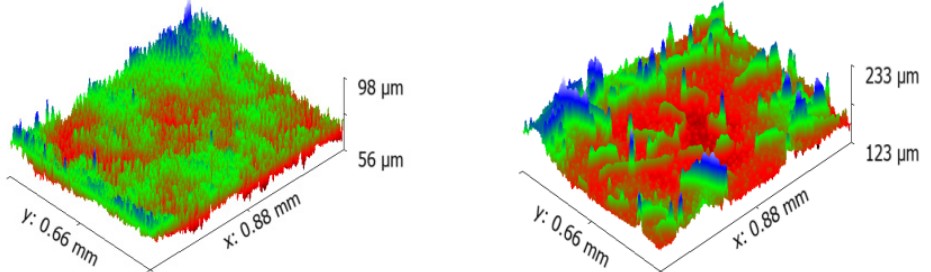

(**a**) Reverse side of the leaf (Longan)     (**b**) Reverse side of the leaf (*Psidium guajava L.*)

**Figure 9.** Target-leaf surface 3D structures for rough factors greater than 1.5.

The results for the target leaf surface 3D structures for rough factors in the range of 1.3–1.5 are shown in Figure 8. The obverse of the *Psidium guajava L.* leaf had many vein dents and some prickly hairs. The rest of the leaves were covered with dense micro-cylinders, and the leaves had no obvious undulations. The reverse of the Litchi leaf surface presented an intensive polygonal structure and widespread dense micro-cylinders, and it was rough to the touch.

Figure 9 shows the target leaf surface 3D structures for rough factors greater than 1.5. The reverse side of the Longan leaf was similar to the reverse side of the litchi leaf. The leaf surface had a dense, tiny polygonal structure with micro-cylinders. However, the micro-cylinders of the Longan leaf surface were more sparse and rougher than those of the Litchi leaf. The obverse of the *Psidium guajava L.* leaf was covered with hook-hairs, the hook-hairs were lodged and the veins were obvious.

In conclusion, the target leaf surfaces with roughness ratios between 1.0~1.3 were slightly undulating and not flat, but the leaf surfaces were relatively flat. When the surface roughness factor was small, the touch was smoother. The target leaf surfaces with rough factors greater than 1.5 generally had dense, fine polygonal structures and were full of little convex areas. When the little convex areas densely covered the leaf to a certain extent, the leaf surface rough factor was reduced to 1.5 or less. In addition, the dense, soft, and thorny leaf surface had a relatively large surface rough factor and a rough touch. Therefore, the rough factors of the target leaf surface between the above two examples were 1.3~1.5.

### 3.1.2. Contact Angle

The contact angle not only represents the performance of the leaf surface but also provides information regarding the interactions between droplets and the leaf surface. The wettability of the leaf surface can be measured using the contact angle.

The measured results of the static contact angle are shown in Figure 10 and Table 9. For the obverse side of the leaf, the contact angle of the *Psidium guajava L.* leaf surface was less than 90°, indicating that the leaf surface is hydrophilic and that liquid can easily moisten the leaf surface. Therefore, the droplet will move more easily on the leaf. The reverse sides of the Litchi and Longan leaves are all hydrophilic.

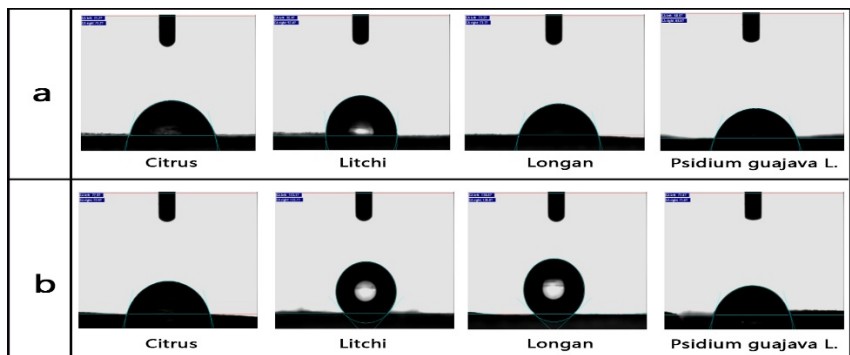

**Figure 10.** Contact angle analysis diagram (**a**) The contact angles of the obverse sides of the leaves; (**b**) The contact angles of the obverse sides of the leaves.

**Table 9.** The contact angles of different leaves.

| Leaf Surface | Plants | | | |
|---|---|---|---|---|
| | Citrus | Litchi | Longan | *Psidium guajava* L. |
| Obverse sides of leaves | 73.4 ± 3.7a | 95.4 ± 2.5b | 72.9 ± 6.4b | 63.1 ± 5.6b |
| Reverse sides of leaves | 76.1 ± 5.2a | 130.5 ± 4.3a | 134.8 ± 4.3a | 82.5 ± 7.2a |

Note: The values are presented as averages ± standard deviation.

Significant differences between means within columns were tested using Duncan's multiple range test at $\alpha$ = 0.05. Different letters denote statistically significant differences.

### 3.1.3. Critical Sliding Volume

The droplet deposition state corresponding to each target leaf surface is shown in Figure 11, and the droplet sliding critical volumes are shown in Table 10.

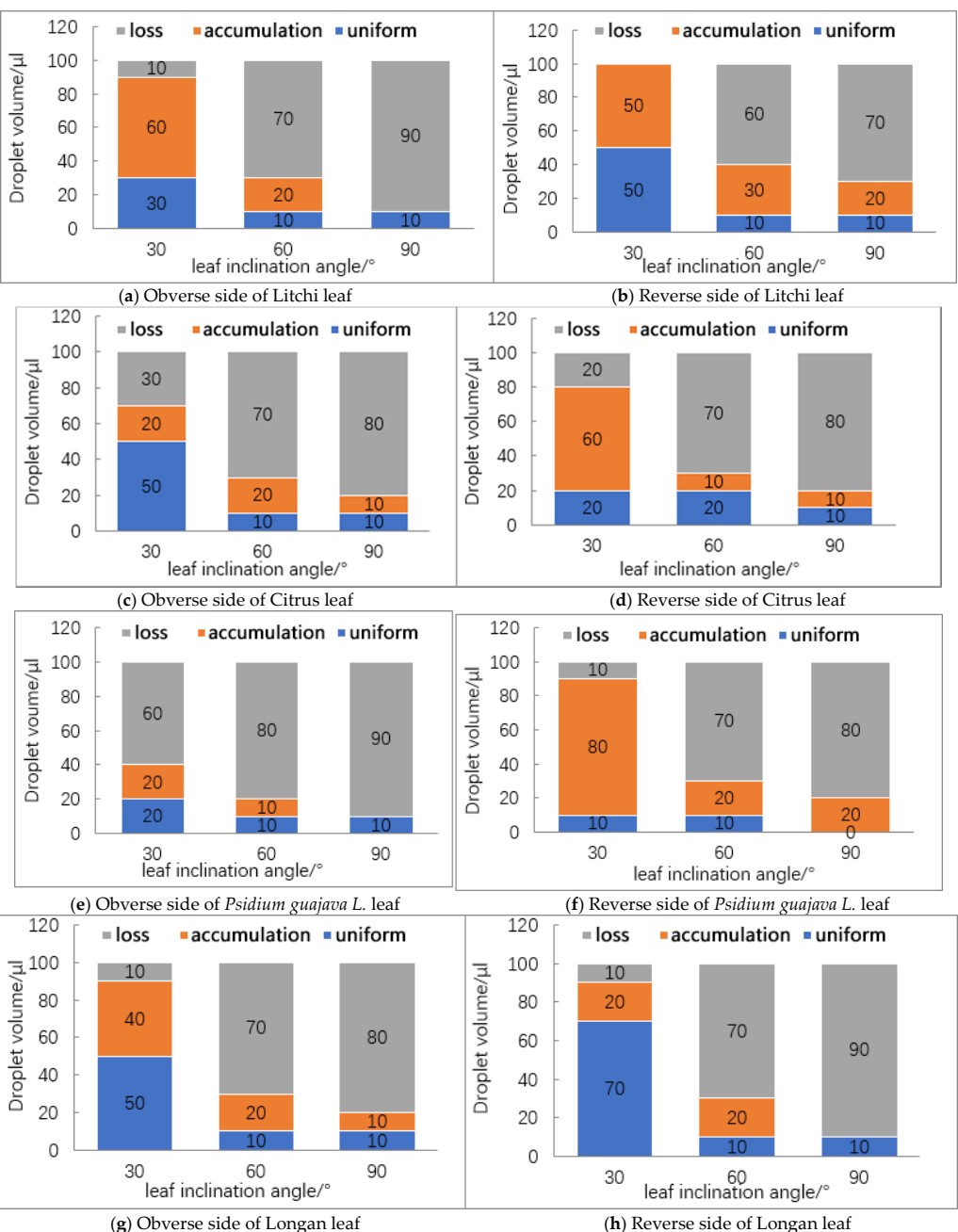

**Figure 11.** Deposition states of liquid droplets on the target leaf surfaces; (**a**) Obverse side of Litchi leaf; (**b**) Reverse side of Litchi leaf; (**c**) Obverse side of Citrus leaf; (**d**) Reverse side of Citrus leaf; (**e**) Obverse side of *Psidium guajava L.* leaf; (**f**) Reverse side of *Psidium guajava L.* leaf; (**g**) Obverse side of Longan leaf; (**h**) Reverse side of Longan leaf.

**Table 10.** The critical sliding volumes of the target droplets.

| Leaf Surfaces | Leaf Inclination Angle | Plant Droplet Sliding Critical Volume/μL | | | |
|---|---|---|---|---|---|
| | | Citrus | Litchi | Longan | *Psidium guajava* L. |
| Obverse of leaf | 30 | 70 ± 0.7 | 90 ± 0.9 | 90 ± 0.9 | 40 ± 0.4 |
| | 60 | 30 ±0.3 | 30 ± 0.3 | 30 ± 0.3 | 20 ± 0.2 |
| | 90 | 20 ± 0.2 | 10 ± 0.1 | 10 ± 0.1 | 10 ± 0.1 |
| Reverse of leaf | 30 | 80 ± 0.8 | 100 ± 2 | 90 ± 0.9 | 90 ± 0.9 |
| | 60 | 30 ± 0.3 | 40 ± 0.4 | 30 ± 0.3 | 30 ± 0.3 |
| | 90 | 20 ± 0.2 | 30 ± 0.3 | 10 ± 0.1 | 20 ± 0.2 |

Note: The measured critical sliding volumes consist of the mean of three repeated critical sliding volumes and the standard deviation.

### 3.2. Discussion of the Adhesion Work Model

(1)  The relationships between leaf characteristics and the droplet deposition state

It can be seen from the fitting formula (Equation (3)) that the leaf adhesion work is proportional to the leaf contact angle and has a quadratic-function relationship with the leaf rough factor. In the range of rough factors from 1 to 1.6, the adhesion work increases with the rough factor. However, when the rough factor is greater than 1.6, the adhesion work begins to decrease.

(2)  The relationship between the droplet diameter and the droplet deposition state

It can be seen from the fitting formula (Equation (3)) that when the leaf angle is constant, the gravitational component of a droplet along an inclined direction increases as the volume of the droplet on the target surface increases. As there is no wind, the kinetic energy of the droplet is zero. When the gravitational component of the droplet is greater than the foliar adhesion, the droplets are gradually wetted in a uniform distribution or accumulation, and the critical volumes of the droplets on each target leaf surface are different.

(3)  The relationship between the leaf inclination angle and the spray droplet deposition state

Multiple linear regression analysis (Equation (3)) suggested that the leaf angle is inversely proportional to the slide work of the hydrophobic leaf. The gravitational component of a droplet along an inclined direction increases with an increase in the target-leaf angle when the droplet volume is constant. Therefore, when the droplet volume is constant, the trend of droplet sliding on the leaf surface can increase with an increase in the leaf inclination angle (0–90).

(4)  Effect of droplet sliding traces on droplet sliding

Additionally, this experiment found that leaf surface sliding traces can aggravate the sliding of leaf surface droplets. For a hydrophobic leaf surface with a large contact angle between the droplet and the leaf surface (such as Litchi and Longan), a sliding droplet produces no sliding trace, and the droplet falls directly off the leaf surface.

### 3.3. Target Leaf Adhesion Work Model Establishment

After the smooth line scatter graph visualization processing of droplet coverage $C_{dro}$ and $d/D^*$ value in Tables 1–4, Figure 12 be obtained. It was found that although the droplet coverage $C_{dro}$ and $d/D^*$ value on different surface properties were different, there was a certain linear fitting relationship between the droplet coverage and $\mathbf{d}/\mathbf{D}^*$ value of four different surface properties. The discovery of this rule provides useful information on the study of the deposition model.

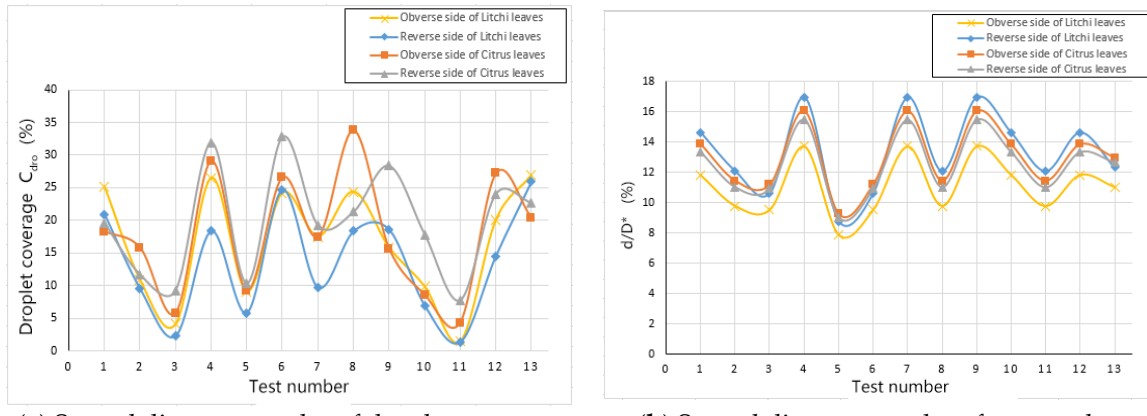

(**a**) Smooth line scatter plot of droplet coverage     (**b**) Smooth line scatter plot of $d/D^*$ value

**Figure 12.** Smooth line scatter plot of spray test data. For profile numbers see Tables 1–4.

Based on the results of the deposition of the positive and negative droplets on the Litchi and Citrus leaves, the prediction accuracies of the prediction model of the airborne spray target leaf surface and the sprayed droplets are as follows: uniform—87.5%; accumulation—80%; loss—100%. For the uniform interaction and accumulation state, especially when the droplet size of the target leaf surface is small, the image method used in this paper will cause statistical errors for fine droplets, which may affect the prediction accuracy of the interaction between the droplet and the leaf surface. For the loss interaction state, the fog leaf coverage rate is calculated according to the leaf area occupied by the target leaf surface, and the droplet leakage of the target leaf surface will cause the droplet coverage to include calculation errors; thus, the droplet and the leaf surface. The accuracy of the interactive state prediction has an impact on the results. Therefore, the prediction model has a lower prediction accuracy for the uniform and accumulation states than for the loss state.

Under the premise that the target leaf surface droplets do not show accumulation or loss, increasing the coverage of leaf droplets can increase the particle size of the spray droplets d and should reduce the critical sliding particle size of the target droplets $D^*$. By combination with the aerodynamic response characteristics of the leaf, the airflow velocity can be appropriately increased to increase the aerodynamic response speed of the leaf, thereby reducing the critical sliding particle size of the droplet $D^*$. For the hydrophilic leaf surface, adding a spray aid to reduce the surface tension of the spray liquid, thereby reducing the contact angle θ between the droplet and the leaf surface, can also achieve the reduction of the critical sliding particle diameter of the droplet $D^*$ and improve the droplet coverage of the target leaf surface. When α is from 0 to 90°, as α increases, the droplet coverage increases.

## 4. Conclusions

This article describes an investigation of the deposition states of sprayed droplets attached to dynamic target leaves. By integrating the air-assisted spray operation parameters and characteristics of the target fruit leaves, the interaction mechanism of the main factors of air-assisted spraying in droplet deposition was investigated, and the behaviors and energy variations during the interactions of droplets attached to the dynamic target leaves and leaf surface were elucidated. The model for predicting the droplet deposition state using an air-assisted spray and the optimal decision-making model for the parameters of the air-assisted spray operation were constructed to fuse the morphological characteristics of leaves and the surface properties of the leaves. The findings provide a theoretical basis for a new research method for the deposition behaviors of atomized spray droplets.

The main results of this study are as follows:

(1) The surface characteristics of four different target leaves were observed and analyzed, and droplet deposition testing was conducted on the reverse and obverse sides of each target leaf. A relationship model between the foliar adhesion work and foliar rough factor, contact angle,

and initial dip angle of the airborne spray target was constructed. The model has a coefficient of fit of 0.917 and can be used for adhesion work calculations. Further analysis results show that the leaf adhesion work is proportional to the leaf contact angle and quadratic function containing the rough factor. In the range of rough factors from 1.0 to 1.6, the adhesion work increases with the increase of the rough factor. When the rough factor is greater than 1.6, the adhesion work begins to decrease. Additionally, based on the energy conservation relationship during the sliding process of the droplets, the critical sliding particle size model of the leaf droplets was proposed.

(2) Axial flow air spray based on the Box–Behnken Design response surface method was designed. In the test, the droplet coverage was obtained under different droplet sizes, application distances, air delivery speeds, and target leaf surfaces. The minimum kinetic energy of the corresponding deposited droplets was selected as the fitting parameter when the droplets slid on the target leaf, and a regression equation between the droplet adhesion work and the turbulent energy of the deposited droplets was constructed. On this basis, combined with the droplet coverage conditions corresponding to the different deposition states of the droplets, the final prediction model of the airborne spray droplet deposition state was proposed. Upon comparing the actual deposited structures with the model-predicted results, it was found that the prediction accuracies of the three states of uniform, accumulation, and loss were 87.5%, 80%, and 100%, respectively.

**Author Contributions:** Supervision, J.L., Z.Y. and H.L.; Writing—original draft, J.L.; Writing—review & editing, J.L., H.C., Y.M., L.X. and Z.L. All authors have read and agreed to the published version of the manuscript.

**Funding:** This research was supported by the earmarked fund for the National Key R&D Program of China (Accession No. 2018YFD0201100), the Special project of Rural Vitalization Strategy of Guangdong Academy of Agricultural Sciences (Accession No.TS-1-4), and the Guangdong Provincal Modern Agricultural Industry Technology System (Accession No. 2019KJ123).

**Acknowledgments:** This research was supported by the earmarked fund for the National Key R&D Program of China (Accession No. 2018YFD0201100), the Special project of Rural Vitalization Strategy of Guangdong Academy of Agricultural Sciences (Accession No.TS-1-4), and the Guangdong Provincal Modern Agricultural Industry Technology System (Accession No. 2019KJ123).

**Conflicts of Interest:** The authors declare no conflict of interest.

## Nomenclature

| | |
|---|---|
| $C_{dro}$ | Droplet coverage (%) |
| $d$ | Diameter of the spray droplet ($10^{-4}$ m) |
| $D$ | Diameter of the deposited droplet ($10^{-4}$ m) |
| $S$ | Spray distance (m) |
| $D^*$ | Critical diameter of the droplet slide on the leaf surface ($10^{-4}$ m) |
| $E_k$ | The droplet instantaneous kinetic energy of landing ($\times 10^{-8}$ J) |
| $E_p$ | Gravitational potential energy of the droplet ($\times 10^{-8}$ J) |
| $E_{dyn}$ | Kinetic energy of the droplet on the leaf ($\times 10^{-8}$ J) |
| $E_{com}$ | Total work ($\times 10^{-8}$ J) |
| $h$ | Height of the leaf from the ground (m) |
| $i$ | Droplet number at one point |
| $m$ | Droplet weight(g) |
| $r$ | Rough factor |
| $V$ | Droplet volume (μL) |
| $v$ | Value of the target-leaf aerodynamic velocity (m·s$^{-1}$) |
| $W_s$ | Adhesion work ($\times 10^{-8}$ J) |
| $\alpha$ | Leaf inclination angle (°) |
| $\rho$ | Droplet density (kg·m$^{-3}$) |
| $\theta$ | Static contact angle (°) |
| $\sigma_{lg}$ | Surface tension of the liquid (N·m$^{-1}$) |
| $m_0$ | The mass of the drop as it falls to the ground (g) |
| $v_0$ | The instantaneous velocity of the drop as it falls to the ground |

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
