# Peer review of "Orchard Spray Study: A Prediction Model of Droplet Deposition States on Leaf Surfaces"

_agronomy, doi:10.3390/agronomy10050747_

Round 1
Reviewer 1 Report
- Instead of working with static contact angle, using contact angle hysteresis makes more sense in this topic.
- l believe, really poor literature review has been done in both the topic of modeling a drop motion on rough surfaces and its relation to topics such as agricultural industry. I recommend looking at Varanasi's group works (especially on agriculture)
- The modeling can be improved by looking at literature on drop motion on rough surfaces
Author Response
Thank you very much for your careful review and valuable Suggestions
, the response to your comments/suggestions is as follows:
1、The static contact Angle measured in this paper reflects the hydrophilic and hydrophobic properties of the leaf surface from another aspect. The maximum adhesion work of a droplet with the same physical property on a leaf with the same physical property will not change. The adhesion work model constructed by multiple linear regression in our paper is actually the maximum adhesion work. The prediction model assumes that if the kinetic energy is not greater than the adhesive work, the droplet will not move.
2、Your suggestion is good, but I can't find the article about agriculture from Varanasi's group, it would be better if you could provide a link to the article.

Reviewer 2 Report
My comments/suggestions for improving the article are the following:
- The rough factor was defined as the ratio between the 'actual' leaf area and the projected leaf area. However, the 'actual' leaf area depends on the resolution of the measurements with which it is measured; a smaller resolution will most likely lead to a higher rough factor. Therefore, please specify the resolution used in the measurements with the profilometer and include measurements of the same surface but with different resolutions in order to study the differences.
- The authors seem to be unaware of the phenomena of contact line pinning. Any contact line moving over a rough surface will experience pinning/depinning events. The level of surface roughness determines how strong the pinning of the contact line is. The authors could attempt to determine/calculate the maximum droplet size that will remain stagnant on a leaf surface due to the pinning of the contact line.
- I strongly encourage the authors to visualize to data contained in table 7, 8, 9 10 with some graphs or any other infographic. The tables can perhaps be included in an appendix.
Author Response
Thank you very much for your careful review and valuable Suggestions
, the response to your comments/suggestions is as follows:
1、The experimental method has been mentioned that ‘the 3D structure of the leaf surface was observed under a 20×0.45 lens’(In section2.2.2). For your proposal to observe leaf roughness at different resolutions, this is a very rigorous way to verify the accuracy of the roughness factor. But due to the epidemic situation, the verification experiment cannot be carried out for the time being.
2、The phenomenon of pinning is due to the role of surface tension, the adhesion force on the surface will lead to the pinning of droplets, thus preventing droplets from sliding on the surface. There is a relationship between nail strength and surface roughness(Contents from the manuscript of "Adhesion of a water droplet on inclined hydrophilic surface and internal fluidity "). Therefore, a roughness factor is introduced to evaluate the surface roughness of the leaf surface to reflect the pinning strength from the side. So the relation between the roughness factor and critical diameter of the droplet is also involved in the critical sliding particle size model of droplets.
3、That's a good suggestion, so I've visualized the valuable data in tables 7 to 10(Figure 12 of the article).

Reviewer 3 Report
This article investigated the deposition states of sprayed droplets attached to dynamic target leaves, which would be helpful for pest management. However, the writing and organization of this manuscript is poor, and it needs a major revision
- Line 29 and 31, typo error, content should be in past tense.
- The citation format was wrong in this manuscript, please read the author guideline carefully.
- Please ask English native speakers to modify the grammar of this manuscript.
- Figure 1 should appear after the content.
- The figures format was not constant, such as Fig. 1 and Figure 2, please correct them.
- The figure resolution was low, please make sure all the figures had 300 dpi or higher.
- Line 227 and 228, existing errors, please correct them.
- In section 3.2 ‘Discussion of the adhesion work model’, there were only a few results explained which were obtained from the experiments, while many discussions. Please correct them. How the results and discussion connected with section 3.1.
- Again, the table (Tables 7 and 8) also needed to follow up with the content
Author Response
Thank you very much for your careful review and valuable Suggestions. The point 1~6 have been modified.
For the point 8, the discussion of the model that you mentioned in section 3.2’Discussion of the adhesion work model’, was based on the model relationship of Eq.3. The content of section 3.1 is a variety of data sets obtained through experiments, which provides data basis for the construction of multiple linear regression model.
For the point 9, a summary of the table data is already available at the end of tables 7 ~10.

Round 2
Reviewer 1 Report
--
Reviewer 2 Report
I think the manuscript in its current form is suitable for publication.
Reviewer 3 Report
My questions were answered well.